# Attenuation of Polycyclic Aromatic Hydrocarbon (PAH)-Induced Carcinogenesis and Tumorigenesis by Omega-3 Fatty Acids in Mice In Vivo

**DOI:** 10.3390/ijms25073781

**Published:** 2024-03-28

**Authors:** Guobin Xia, Guodong Zhou, Weiwu Jiang, Chun Chu, Lihua Wang, Bhagavatula Moorthy

**Affiliations:** 1Section of Neonatology, Department of Pediatrics, Baylor College of Medicine and Texas Childrens’ Hospital, Houston, TX 77030, USA; guobin.xia@bcm.edu (G.X.); weiwuj@bcm.edu (W.J.); chunc@bcm.edu (C.C.); lihuaw@bcm.edu (L.W.); 2Institute of Biosciences and Technology, College of Medicine, Texas A&M University, Houston, TX 77030, USA

**Keywords:** polycyclic aromatic hydrocarbons (PAHs), ^32^P-postlabeling assay, DNA adducts, omega-3 fatty acids, cytochrome P450 1 enzymes, CYP1B1, carcinogenesis, epigenetic regulation

## Abstract

Lung cancer is the leading cause of cancer death worldwide. Polycyclic aromatic hydrocarbons (PAHs) are metabolized by the cytochrome P450 (CYP)1A and 1B1 to DNA-reactive metabolites, which could lead to mutations in critical genes, eventually resulting in cancer. Omega-3 fatty acids, such as eicosapentaenoic acid (EPA) and docosahexaenoic acid (DHA), are beneficial against cancers. In this investigation, we elucidated the mechanisms by which omega-3 fatty acids EPA and DHA will attenuate PAH-DNA adducts and lung carcinogenesis and tumorigenesis mediated by the PAHs BP and MC. Adult wild-type (WT) (A/J) mice, *Cyp1a1*-null, *Cyp1a2*-null, or *Cyp1b1*-null mice were exposed to PAHs benzo[a]pyrene (BP) or 3-methylcholanthrene (MC), and the effects of omega-3 fatty acid on PAH-mediated lung carcinogenesis and tumorigenesis were studied. The major findings were as follows: (i) omega-3 fatty acids significantly decreased PAH-DNA adducts in the lungs of each of the genotypes studied; (ii) decreases in PAH-DNA adduct levels by EPA/DHA was in part due to inhibition of CYP1B1; (iii) inhibition of soluble epoxide hydrolase (sEH) enhanced the EPA/DHA-mediated prevention of pulmonary carcinogenesis; and (iv) EPA/DHA attenuated PAH-mediated carcinogenesis in part by epigenetic mechanisms. Taken together, our results suggest that omega-3 fatty acids have the potential to be developed as cancer chemo-preventive agents in people.

## 1. Introduction

A total of 238,340 new cases of lung cancer and 127,070 lung cancer deaths are projected to occur in the United States in 2023 [1]. Lung cancer is the leading cause of cancer death globally, with an estimated 1.8 million deaths (18%) in 2020 [2]. Epidemiological, basic, and clinical research data have shown that most lung cancers are caused by environmental factors, including exposure to complex mixtures such as carcinogenic polycyclic aromatic hydrocarbons (PAHs) from cigarette smoke or air pollution [3,4,5]. The primary sources of PAHs are identified as anthropogenic in origin, such as the exhaust of motor vehicles, petroleum refineries, power plants, burning coal, the combustion of refuse, oil/gasoline spills, tobacco smoke, barbeque smoke, and coke production [6]. The main sources of PAH exposure occur in cities, such as vehicles and industries including transportation, the burning of fossil fuels (oil and coal) in generating stations, and factories [7].

Cytochrome P450s (CYP) are heme-containing enzymes which catalyze various phase I metabolic reactions, such as C-, N-, and S-oxidation and dealkylation [8]. CYP enzymes, especially those belonging to the CYP1 family (CYP1A1, 1A2, 1B1), play critical roles in the metabolic activation of PAHs to reactive metabolites that can bind to DNA to form DNA adducts associated with mutagenesis [3,4,9,10,11,12,13,14,15]. If not repaired, these mutated nucleotides can lead to cancers in target and other organs, such as lung, liver, stomach, colon, breast, skin, kidney, etc. [3,4,9,10,11,12,13]. Such pre-mutagenic lesions play essential roles in the initiation phase of carcinogenesis [9,10,11,12,13]. Animal experiments have shown a positive correlation between levels of PAH-DNA adducts at the initial stages and, later, tumor incidence [12,14]. Ceppi et al. [16] reported that bronchial adducts are strongly associated with lung cancer risk in humans based on a meta-analysis involving twenty-two studies for a total of 1091 subjects, 887 lung cancer cases, and 204 apparently healthy individuals with no evidence of lung cancer. Therefore, a decrease in PAH-DNA adduct levels is expected to diminish or reduce the potential development of this type of cancer. The formation of PAH-DNA adducts can be attenuated by detoxification. Mutations induced by DNA adducts can be removed by DNA repair processes [17,18,19], and damaged cells can be eliminated by cell death, including apoptosis [20,21,22]. 

Human CYP1B1 is found mainly in extrahepatic tissues and is overexpressed in a variety of human tumors [23,24,25]. Meta-analysis of clinical studies indicated that a CYP1B1 polymorphism was associated with a wide variety of cancers, including lung cancer, breast cancer, and colon cancer [26,27]. It was also reported that CYP1B1 promotes cell proliferation and metastasis by inducing EMT and Wnt/β-catenin signaling via Sp1 induction [28]. Recently, we reported [29] that omega-3 fatty acids attenuated levels of DNA adducts in part via the inhibition of CYP1B1. It has also been reported [30] that epoxy metabolites of docosahexaenoic acid (DHA) produced by CYP enzymes inhibit angiogenesis, tumor growth, and metastasis. Therefore, CYP1 enzymes may play important roles in the omega-3 fatty acid-mediated prevention of chemical carcinogenesis. 

Exposure to PAHs cause several molecular alterations, such as DNA damage, DNA adduct formation, and genetic aberrations that are involved in epigenetic regulations that are associated with carcinogenesis and tumorigenesis [31]. Epigenetic modifications, such as DNA methylation, and RNA modification have been reported to play an important role in lung cancer development and other pulmonary diseases [32]. It was reported that dietary fish and omega-3 fatty acids were associated with lower ABCA1 DNA methylation levels in a Japanese population [33].

Most cancers are still not curable but are preventable. Natural compounds have been presumed to be safer than synthetic compounds due to their presence in diet, wide availability, and tolerability [34]. ]. Omega-3 fatty acids are beneficial against a variety of medical conditions including cancer prevention [32]. It was reported that the treatment of rats with fish oil containing omega-3 fatty acids eicosapentaenoic acid (EPA, 90 mg) and docosahexaenoic acid (DHA, 60 mg) before administration of the carcinogen 7,12-dimethylbenz(a)anthracene (DMBA) significantly decreased mammary tumor incidence compared with a control [35,36]. Our previous experiments have indicated [29,37] that dietary fish oil (FO) or EPA and DHA decreased PAH-DNA adduct formation in the lungs and livers of mice by 30–45%. We also found that this attenuation of DNA adduct formation by EPA/DHA was, at least in part, through the inhibition of CYP1B1. Therefore, as understanding the mechanisms of carcinogenesis is the basis of cancer prevention, it is very important to clarify how omega-3 fatty acids influence PAH metabolism in order to apply these natural compounds in an anticancer strategy, especially in high-risk populations such as smokers and others exposed to environmental carcinogens.

In this study, we investigated the mechanisms by which the omega-3 fatty acids EPA and DHA elicit the attenuation of PAH-DNA adducts and further decrease lung tumor incidence and/or tumor number in mice treated with the PAHs benzo[a]pyrene (BP) or 3-methylcholanthrene (MC). Understanding the mechanisms involved in the attenuation of chemical carcinogenesis and tumorigenesis by omega-3 fatty acids EPA and DHA should produce highly significant, new qualitative and quantitative data in the area of cancer prevention.

## 2. Results

### 2.1. Dietary Omega-3 Fatty Acids Modulate Levels of Pulmonary PAH-DNA Adducts in WT and Cyp1a1-Null Mice (Experiment 1)

After one month on special omega-3 fatty acid diets, WT, *Cyp1a1*-null, *Cyp1b1*-null, and *Cyp1a2*-null mice were treated with MC and were then terminated 7 days after MC treatment. Typical patterns of bulky DNA adducts were exhibited in all four genotype mice treated with MC (Figure 1A). These adducts, were derived from 7,8-epoxy-7,8,9,10-tetrahydro-(1 or 2), and 9,10-trihdroxy-3-hydroxy-MC, which are the ultimate carcinogenic metabolites, as we have shown earlier [14]. As shown in Figure 1B–E, dietary omega-3 fatty acids FO, EPA, DHA, or EPA + DHA significantly decreased pulmonary PAH-DNA adducts in all four genotypes of mice. Dietary omega-3 fatty acids attenuated levels of DNA adducts in WT mice by about 30% compared to the CO diet (Figure 1B). Interestingly, further attenuation of pulmonary DNA adducts (up to 57.68%) was observed in *Cyp1a1*-null mice (Figure 1C) by dietary omega-3 fatty acids. Furthermore, the lowest levels of lung DNA adducts were observed in *Cyp1b1*-null mice compared with the other three genotypes of mice and the adduct levels were significantly reduced by dietary EPA/DHA (Figure 1D). Interestingly, *Cyp1a2*-null mice displayed the highest levels of pulmonary DNA adducts among the four genotypes of mice (Figure 1E), although CYP1A2 is liver-specific, suggesting that hepatic CYP1A2 plays a key role in the detoxification of MC. Dietary omega-3 fatty acids caused significant suppression of PAH-DNA adduct levels in mice lacking CYP1A2 by about 33–51% compared to the CO diet (Figure 1E).

### 2.2. EPA/DHA Decreases the Levels of PAH-DNA Adducts and Inhibits CYP1B1 Expression

To further understand the relationship between early stage biomarkers (DNA adducts) and late-stage tumor formation and to explore the mechanisms underlying the prevention of PAH-mediated carcinogenesis by omega-3 fatty acids, in vivo experiments at two time points (7 days or 120 days after BP or MC treatment) were carried out. For details, see Section 4.4.2. Typical profiles of ^32^P-postlabeled BP-DNA adducts are displayed in the lungs of mice 7 days after BP treatment (Figure 2A). The major adduct (spot 1) is benzo(a)pyrene-7,8-dihydrodiol-9,10-epoxide deoxy-guanosine (BPDE-dG), based on previous publications [29,37]. Figure 2B shows that pure EPA/DHA significantly decreased the formation of BP-DNA adducts in the lungs and livers of both male and female mice. Pulmonary PAH adducts were diminished by 33% and 61% by EPA/DHA in male and female mice, respectively. Hepatic PAH-DNA adducts were also significantly decreased by dietary EPA/DHA in both male and female mice (Figure 2B). 

We next investigated the mechanisms by which omega-3 fatty acids EPA/DHA attenuated PAH-DNA adducts in mice treated with carcinogenic PAHs. On the basis of the results as shown in Figure 1 and Figure 2, we hypothesized that EPA and DHA suppressed the formation of pulmonary DNA partially by attenuation of PAH bioactivation via CYP1B1 inhibition. Western blot analysis supported our hypothesis that DHA significantly inhibited the expression of lung CYP1B1 by approximately 40%, while EPA has almost 20% CYP1B1 inhibition (Figure 2C). It is noteworthy that CYP1B1 is reported to play a key role in the bio-activation of BP to DNA-binding metabolites [38]. 

### 2.3. EPA/DHA Decreases the Formation of Lung Tumors in A/J Mice

A/J mice received EPA (60 mg/kg) and DHA (40 mg/kg) by oral gavage from day 1 to day 21. Animals were terminated 120 days after BP or MC treatment (For details, see Section 4.4.2). Pulmonary tumor incidence induced by BP was significantly decreased by EPA/DHA (63.1%) compared to the CO group (94.7%), with the *p*-value being 0.017 (Figure 3A). Correspondingly, 100% lung tumor incidence was observed in both the CO and EPA/DHA groups of mice treated with MC. However, tumor colony numbers in the lungs were significantly diminished by about 43% by EPA/DHA (6.69 ± 0.61) compared to CO (11.77 ± 1.71), *p* < 0.01. The tumor number in mice treated with BP was 3.32 ± 0.48 for the CO group and 1.05 ± 0.22 for the EPA/DHA group with a *p*-value < 0.001 (Figure 3B). Histological examination also showed that EPA/DHA markedly suppressed tumorigenesis in the lungs (Figure 3C,D).

### 2.4. Inhibition of sEH Enhances EPA/DHA-Mediated Prevention of Pulmonary Carcinogenesis

Typical patterns of pulmonary BP adducts were observed (Figure 2A). Mice treated with CO, EPA/DHA, EPA/DHA + t-TUCB, or TPPU [39] showed similar profiles of DNA adducts, indicating that sEH inhibitor t-TUCB (1.5 mg/kg) or TPPU (1.5 mg/kg) did not induce additional DNA adducts. Furthermore, EPA/DHA by themselves attenuated BP-DNA adducts, and this effect was further decreased in the presence of the sEH inhibitors, suggesting that higher levels of epoxy omega-3 fatty acids suppressed the formation of DNA adducts (Figure 4).

### 2.5. Attenuation of Expression of PAH-Induced Genes Associated with Epigenetic Regulation by EPA/DHA In Vivo

We tested the hypothesis that EPA/DHA suppressed PAH-mediated carcinogenesis in part by inhibiting genes associated with epigenetic regulation (DNMT3a, EZH2, and onco miRs 17 and 19-b1) that are known to be involved in tumorigenesis and by enhancing the expression of RUNX3, a tumor suppressor gene. We used the same tissues that we used for DNA adduct studies (shown in Figure 2A,B) to study the effect of PAHs in the presence and absence of EPA/DHA, as seen in Figure 5A–D. The PAHs BP and MC induced the expression of DNMT3a, EZH2, miR-17, and miR19b-1 in the lungs at 7 days, and this induction was suppressed in mice that also received EPA/DHA (Figure 5A–D). In contrast, EPA/DHA augmented the expression of RUNX3 compared to its level in mice treated with MC (Figure 5E).

Based on the above results, we hypothesized that PAHs cause silencing of RUNX3 by adding methyl groups in CpG islands in the RUNX3 promoter via DNMT1/3a and by enhancing the EZH2-mediated trimethylation of H3K27me3, which in turn leads to the suppression of RUNX3 (Figure 6). Thus, our data demonstrate the suppression of DNMT3a and EZH2 and the induction of RUNX3 by EPA/DHA (Figure 5A,B,E). These results support the hypothesis that EPA/DHA protects against PAH-mediated lung tumorigenesis by decreasing the expression of DNMT and EZH2, which in turn leads to the reversal of RUNX3 silencing and, therefore, results in the increased expression of RUNX3. 

## 3. Discussion

Carcinogenic PAHs can be metabolized by CYP1A1 and 1B1 to form DNA adducts, which can lead to mutations and the initiation of carcinogenesis [40,41,42]. PAH exposure also leads to the upregulation of CYP1A1 transcription by binding to the aryl hydrocarbon receptor (AHR) and eliciting the transcription of the CYP1A1 promoter, which comprises specific xenobiotic-responsive elements [3]. MC induced the lowest levels of DNA adducts in *Cyp1b1*-null mice (Figure 1D) compared to those in WT mice (Figure 1B), indicating that CYP1B1 plays an important role in bioactivation in PAH metabolism. On the contrary, the highest levels of DNA adducts were observed in *Cyp1a2*-null mice (Figure 1E), suggesting that CYP1A2 plays key role in detoxification in PAH metabolism [15]. Previous research has indicated a significant correlation between DNA adduct levels at an early stage (1, 7, or 21 days after PAH treatment) and mouse mortality as well as the total incidence of tumors, including liver, lung, and forestomach. In this study, we demonstrated that dietary omega-3 fatty acids, EPA/DHA, substantially suppressed the formation of pulmonary DNA adducts induced by MC in WT, *Cyp1a1*-null, *Cyp1b1*-null, and *Cyp1a2*-null mice (Figure 1). Intriguingly, EPA/DHA elicited an even more pronounced attenuation of DNA adduct concentrations in *Cyp1a1*-null (Figure 1C), implying that when CYP1B1 (which was the primary enzyme for PAH bioactivation in the absence of CYP1A1) was inhibited by EPA/DHA, its function in bioactivating MC was markedly diminished as well (Figure 1). We postulate that EPA/DHA induced the downregulation of CYP1B1 potentially through mechanisms involving the CYP1B1 promoter and/or gene methylation [43].

CYP1A1 and 1B1 constitute the principal enzymes responsible for metabolizing PAHs into ultimate carcinogens [3,44]. Previous studies have reported that the CYP1A1 enzyme possess dual functions in both the detoxification and bioactivation pathways of PAH metabolism [45,46]. Additionally, prior evidence indicates that omega-3 fatty acids decrease CYP1A1 enzymatic activity [47]. In this work, we examined the levels of pulmonary and hepatic DNA adducts in both male and female mice following the administration of pure EPA and DHA. We demonstrate that EPA/DHA significantly reduced the levels of DNA adducts in the lungs and livers of A/J mice, in both males and females (Figure 2B). Given that the liver is a vital metabolic organ enriched with additional enzymes involved in PAH detoxification, such as CYP1A2, the hepatic adduct levels were considerably lower compared to the lungs. These hepatic data are critically important and should facilitate elucidation of the mechanisms underlying pulmonary carcinogenesis.

Possible mechanisms through which omega-3 fatty acids attenuate PAH-DNA adducts and inhibit carcinogenesis are as follows. (1) Dietary fish oil containing EPA/DHA may induce the expression of certain CYP enzymes, including CYP1A1 and CYP1A2, while suppressing CYP1B1 (Figure 2C). This modified CYP expression profile could subsequently facilitate the detoxification of PAHs [37]. (2) CYP2C, CYP1A1, and CYP1A2 [48] may catalyze the conversion of omega-3 fatty acids into epoxy metabolites, such as epoxy EPA [49] and epoxy DHA [30]. These epoxy metabolites have been demonstrated to inhibit angiogenesis, tumor growth, and metastasis and may suppress chemical carcinogenesis (Figure 4). (3) EPA/DHA directly inhibits the expression of some genes associated with epigenetic regulation induced by PAH exposure (Figure 5). Epidemiological and preclinical evidence supports the notion that a diet rich in omega-3 dietary fatty acids is associated with reduced risks of several diseases, including heart diseases and cancer [50]. Dietary omega-3 fatty acids may play role in inhibiting AhR-dependent cancer progression [51].

Data from long-term animal tumorigenesis experiments displayed trends analogous to those of the short-term DNA adduct experiments. EPA/DHA profoundly suppressed the incidence of pulmonary tumors induced by BP (Figure 3A). Moreover, lung tumor number elicited by BP and MC was markedly reduced upon EPA/DHA administration (Figure 3B). CYP1B1 may play a pivotal role in both DNA adduct formation (short-term) and tumorigenesis (long-term). Previous studies have reported that microRNA-187-5p attenuates cancer cell progression in non-small cell lung cancer through the transcriptional repression of CYP1B1 [52]. Additionally, CYP1B1 has been shown to potentiate cell proliferation and metastasis via the induction of EMT and the activation of Wnt/β-Catenin signaling mediated by SP1 upregulation [28]. Independent work by Lin, et al. demonstrated that MECP2 protects against cigarette smoke extract-induced lung epithelial cell injury potentially by downregulating CYP1B1 expression through increased CYP1B1 promoter methylation [53]. Furthermore, Li, et al. observed a distinct increase in CYP1B1 promoter CpG island methylation in isoniazid-induced hepatotoxicity in rodents [54]. Meta-analyses of clinical investigations have indicated that polymorphic variants of CYP1B1 associate with diverse malignancies including lung, breast, and colon cancers [26]. Additionally, Li, et al. reported that CYP1B1 inhibition could contribute to reduced tumorigenesis [55]. Collectively, these data suggest EPA/DHA may hinder pulmonary carcinogen-induced carcinogenesis and tumorigenesis partly via the suppression of CYP1B1 activity.

Omega-3 fatty acids can be metabolized by CYP enzymes into epoxy omega-3 derivatives, i.e., EEQ and EDP, that will inhibit tumor growth and metastasis by suppressing tumor angiogenesis [30]. However, epoxy omega-3 EEQ/EDP are unstable and will be further hydrolyzed into inactive diols by sEH [56]. sEH is a cytosolic enzyme that catalyzes the rapid hydration of EEQ and EDP by adding water to these epoxygenated fatty acids (EpFAs) and converting them into inactive or less active 1,2-diols [56,57]. Xia et al. reported [58] that a combination of use omega-3 fatty acids and sEH inhibition is a strategy with high potential for pancreatic cancer treatment and prevention. It was reported that the co-administration of EDP with low doses of the sEH inhibitor t-AUCB yielded an approximate 2-fold increase in the EDP level in circulation [59]. It has been also reported that sEH inhibitors block EEQ and EDP to be further converted to their inactive diols [60]. Recently, it was also reported [61] that sEH and epoxy fatty acids play a vital role in inflammation and protect the lungs form environmental PM2.5 particulate pollution. Therefore, mice deficient in sEH or WT mice treated with sEH inhibitors t-TUCB or TPPUB that are maintained on the EPA/DHA diet will metabolize EPA/DHA into their epoxy derivatives by CYP1A2 [30,59,62,63] or CYP2C [30,59,63,64,65], resulting in higher levels of EEQ and EDP. The presence of EEQ and EDP in the serum or tissues will be stabilized due to a lack of sEH, resulting in the blockade of sEH-mediated metabolism [59]. TPPU is a dual inhibitor for sEH and COX-2 [66]. As indicated in Figure 4, EPA/DHA significantly decreased BP adduct levels compared to the CO group. Profoundly lower levels of BP DNA adducts were observed in mice treated with a combination of EPA/DHA and the sEH inhibitors t-TUCB or TPPU, indicating that higher levels of epoxy omega-3 fatty acids further suppressed the formation of DNA adducts. 

The microRNAs miR-17 and miR-19b are components of the oncogenic miR-17-92 cluster, which exhibits overexpression in human lung cancers [67]. The epigenetic regulators EZH2 and DNMT3a as well as the oncomiRs miR-17 and miR-19b have established roles in tumorigenesis [68,69]. We demonstrate that these tumorigenic genes associated with epigenetic regulation are induced by carcinogenic PAH exposure but are inhibited upon treatment with EPA/DHA (Figure 5A–D), potentially via enhancing the expression of the tumor suppressor gene RUNX3 (Figure 5E and Figure 6) [70,71]. EZH2 is overexpressed in numerous cancers, including lung cancer, where it appears to promote oncogenesis through silencing transcription (via trimethylation of H3K27) and blocking differentiation [72]. Prior reports indicate associations between EZH2 and cancer initiation, progression, metastasis, metabolism, drug resistance, and immune regulation [73]. Our results revealed the EPA/DHA-mediated suppression of EZH2 expression (Figure 5E). DNA methyltransferases (DNMTs) can contribute to cancer development through the hypermethylation-induced silencing of tumor suppressor genes [74]. Runt-related transcription factor 3 (RUNX3) is one such tumor suppressor [75] whose inactivation is considered to be an early pivotal event in lung adenocarcinoma pathogenesis [70]. Collectively, these data suggest that the downregulation of pro-tumorigenic epigenetic regulators may partially mediate the chemo-preventive effects of EPA/DHA (Figure 6). Supporting this hypothesis, studies by Fujii, et al. [76], Kodach, et al. [77], and Ciavatta, et al. [78] demonstrate that RUNX3 repression in human cancers and diseases such as vasculitis enables increased tumorigenesis.

Given that many cancers remain incurable, it is crucial to understand the mechanisms underlying chemical carcinogenesis for effective cancer prevention. Lung cancer is a significant global health concern, as it is the leading cause of cancer-related deaths worldwide [1,79]. Prevention of PAH-induced lung carcinogenesis may have significant implications for the development of chemo-preventive strategies for lung cancer.

## 4. Materials and Methods

### 4.1. Chemicals

BP was obtained from Sigma-Aldrich (St. Louis, MO, USA). MC was obtained from Toronto Research Chemicals (Cat No. M294460, Toronto, ON, USA). Fish oil (from menhaden) was also purchased from Sigma-Aldrich (St. Louis, MO, USA). Omega-3 fatty acids EPA and DHA were purchased from Cayman Chemical (Ann Arbor, MI, USA). sEH inhibitors Trans-4-[4-(3-adamantan-1-yl-ureido)-cyclohexyloxy]-benzoic acid (t-TUCB) and 1-trifluoromethoxyphenyl-3-(1-propionylpiperidin-4-yl) urea (TPPU) were kindly provided by Dr. Bruce Hammock (UC, Davis). CYP1B1 antibody was generously provided by Dr. Colin Jefcoate (University of Wisconsin-Madison). GRP78 polyclonal antibodies were purchased from Abcam (Cat. No. ab21685, Waltham, MA, USA). Materials for DNA extraction [80,81,82,83], RNA extraction [84], and ^32^P-postlabeling analysis [80,85,86] have been reported previously. 

### 4.2. Animals

A/J mice were obtained from The Jackson Laboratory (Bar Harbor, ME, USA). *Cyp1a1*-null, *Cyp1b1*-null, and *Cyp1a2*-null mice (A/J) were generated by backcrossing C57BL/6J/Sv129 *Cyp1a1*-null mice with wide-type A/J mice for 10 generations. The *Cyp1a1*-null mouse breeding pairs were generously donated by Dr. Daniel Nebert (University of Cincinnati). *Cyp1a2*-null and *Cyp1b1*-null breeding pairs were obtained from Dr. Frank Gonzalez (NCI, Bethesda, MD, USA). All animal experiments adhered to the principles outlined in the Guide for the Care and Use of Laboratory Animals, as endorsed by the U.S. National Institutes of Health. The animal protocol #AN-907 was approved by the Animal Experimental Ethical Committee of Baylor College of Medicine.

### 4.3. Diet

Diets containing corn oil (CO), fish oil (FO), EPA, DHA, and EPA + DHA were made by Dytes Inc. (Bethlehem, PA, USA). Dietary EPA and DHA were purchased from Nu-Chek Prep, Inc. (Elysian, MN, USA). All diets contained 5% lipid by weight. The composition of each diet is listed in Table 1 [29]. To prevent formation of oxidized lipids, the diets were stored at −20 °C in the dark. The omega-3 fatty acid diets contained 1 g/kg γ-tocopherol and 0.025% tertiarybutylhydroquinine (TBHQ) as antioxidants [87]. Food-grade corn oil was also supplemented with α-tocopherol, γ-tocopherol, and TBHQ to obtain antioxidant levels equivalent to those in the omega-3 fatty acid diets.

### 4.4. Animal Experiments

#### 4.4.1. Experiment 1: Effects of Dietary Omega-3 Fatty Acids on Formation of DNA Adducts

Male WT, *Cyp1a1*-null mice, *Cyp1a2*-null, and *Cyp1b1*-null (all on A/J background (6–8 wk.)) were used in this experiment. After one month on special corn oil (CO) (5%), FO (5%), EPA (750 mg/kg), DHA (500 mg/kg), or EPA + DHA (750 + 500 mg/kg) diets, animals were treated with a single dose of MC (25 mg/kg) by intraperitoneal (i.p.) injection. Animals were maintained on the same diet until termination at 7 days after MC treatment.

#### 4.4.2. Experiment 2: Effects of EPA/DHA on Lung DNA Adducts and Tumor Formation

EPA (60 mg/kg) and DHA (40 mg/kg) were dissolved in CO with volume of 6.7 µL per gram body weight. A/J mice (male and female, 8 wk.) received EPA and DHA by oral gavage from day 1 to day 21. Control mice received the same volume of CO (6.7 µL/g b.w.). Animals were maintained on AIN-76 semifurified diet that was obtained from MP Biomedicals (Santa Ana, CA, USA). On day 3, mice were treated with BP (10 mg/kg for DNA adduct studies, 60 mg/kg for tumorigenesis) or MC (10 mg/kg) by i.p. injection. For DNA adduct studies, 4 mice per group were terminated on day 10 (7 days after BP or MC treatment). For the tumorigenesis experiments, mice continued to receive EPA/DHA twice per week until the end of the experiment (120 days). Each group had at least 20 animals (10 male and 10 female). Tumor formation was determined by visual examination (tumor number) and histological analyses were conducted by a pathologist who provided his findings in a blinded fashion.

#### 4.4.3. Experiment 3: Inhibition of CYP1B1 Expression by EPA and DHA

Male A/J mice (8–10 wk., 4 mice per group) were treated with EPA (100 mg/kg) and/or DHA (60 mg/kg) by gavage for 3 days. EPA and DHA were dissolved in CO with volume of 6.7 µL per gram body weight. Mice were fed the AIN-76 semipurified diet. On day 2, the mice treated i.p. with BP (25 mg/kg). The mice were euthanized 48 h after BP treatment, lung tissues were collected, and lung microsomes were isolated by differential centrifugation. Western blot analysis was conducted to determine CYP1B1 protein levels by using GRP78 as a loading control. Blots were quantified with BioRad ChemiDoc Touch and Image Lab software (Image lab version 6.1) (BioRad Laboratories, Hercules, CA, USA).

#### 4.4.4. Experiment 4: Inhibition of sEH Enhances EPA/DHA-Mediated Prevention of BP-Induced Lung Carcinogenesis

Male A/J mice (8 wk.), were pre-treated with EPA (60 mg/kg)/DHA (40 mg/kg), EPA (60 mg/kg)/DHA (40 mg/kg) + sEH inhibitor t-TUCB (1.5 mg/kg), or EPA (60 mg/kg)/DHA (40 mg/kg) + TPPU (1.5 mg/kg) for three days, and then animals were treated with BP (60 mg/kg) by i.p. Mice continued to receive EPA/DHA or EPA/DHA + sEH inhibitor for an additional day. During the experiment, the animals were maintained on AIN-76 semipurified diet. Animals were terminated on day 5 (2 days after BP treatment).

All animal studies were approved by the IACUC of Baylor College of Medicine.

### 4.5. DNA Adduct Analysis

DNA extraction and ^32^P-postlabeling analyses were performed as in previous experiments [12,37]. DNA was isolated by solvent extraction combined with enzymatic digestion of protein and RNA [82,83] and stored at −80 °C until analysis. The nuclease P1-enhanced bisphosphate version of the ^32^P-postlabeling method [80] was used for analysis, with modifications of the chromatographic (TLC) conditions [88]. Briefly, DNA (10 μg) was enzymatically degraded to normal (Np) and modified (Xp) deoxyribonucleoside 3′-monophosphates with micrococcal nuclease and spleen phosphodiesterase at pH 6.0 and at 37 °C for 3.5 h. After treatment of the mixture with nuclease P1 to convert normal nucleotides to nucleosides, modified nucleotides (Xp) were converted to 5′-^32^P-labeled deoxyribonucleoside 3′,5′-bisphosphates (pXp) by incubation with carrier-free [γ-^32^P] ATP and polynucleotide kinase. Radioactively labeled digests were applied to modified PEI-cellulose thin layers and chromatographed overnight (15–16 h) with solvent 1 (D1, 2.3 M sodium phosphate, pH 5.70) to purify bulky adducts. Labeled PAH adducts retained in the lower (L, 2.8 × 1.0 cm) part of the D1 chromatogram were each, after brief autoradiography on Cronex 4 X-ray film, contact-transferred to individual acceptor sheets and resolved by two-dimensional TLC. The bulky DNA adducts were separated with solvents 2 (3.82 M lithium formate, 6.75 M urea, pH 3.35) and solvent 3 (0.72 M sodium phosphate, 0.45 M Tris-HCl, 7.65 M urea, pH 8.20) in the first and second dimensions, respectively. ^32^P-labeled DNA adducts were visualized by screen-enhanced autoradiography at −80 °C using Kodak XAR-5 film and were detected with the aid of Amersham Typhoon Biomolecular Phosphorimager (GE Healthcare, Chicago, IL, USA). Appropriate blank count rates were automatically subtracted by the instrument from sample values. The extent of covalent DNA modification was calculated from corrected sample count rates. Levels of DNA adducts were calculated as previously reported [86]. Quantitative data represented minimum estimates because 100% recovery presumably was not achieved. 

### 4.6. qRT-PCR Analysis

Total RNA was extracted from lung tissue using Direct-zol RNA Miniprep kit (R2052, ZYMO Research, Irvine, CA, USA). RNA concentration and absorbance ratios (A_260/280_ and A_260/230_) were measured by spectrophotometer, DS-11 Spectrophotometer (DeNovix Inc., Wilmington, DE, USA). The cDNA was prepared using the iScript cDNA Synthesis kit (Cat. No. 1708890, Bio-Rad, Hercules, CA, USA). QPCR was performed using the Quant Studio 7Pro real-time PCR detection system (Applied Biosystem, Waltham, MA, USA) and SYBR Green Supermix (Cat. No. 1708880, Bio-Rad). Thermal cycling conditions were used based on the manufacturer’s instructions. The primers used in the real-time PCR test are listed in Table 2. All primers were made by Sigma. Relative mRNA levels were calculated using the 2^−ΔΔCT^ method and normalized to B-Actin expression, which was used as the housekeeping gene.

### 4.7. Histological Examination

The animals were euthanized by CO_2_ asphyxiation and cervical dislocation. The livers were removed for various analyses. Part of each tissue was fixed in 4% paraformaldehyde, following which the samples were embedded in paraffin and cut into 5 µm thick sections. The tissue section on the glass slide was stained with hematoxylin and eosin (H&E).

### 4.8. Statistical Analysis

Statistical analysis of the data was performed by Student’s t-test or one-way analysis of variance (ANOVA) with multiple comparison by Newman–Keuls test [89].

## 5. Conclusions

Our studies demonstrate that omega-3 fatty acids can significantly attenuate lung DNA adduct formation and tumorigenesis in mice exposed to carcinogenic PAHs through multiple molecular mechanisms, including decreasing PAH-DNA adducts, downregulating CYP1B1 expression, inhibiting epigenetic genes, and protecting against lung tumorigenesis via the RUNX3 gene. These results suggest that omega-3 fatty acids may be a promising chemo-preventive agent for lung cancer in high-risk populations. Future studies should explore the efficacy and safety of supplemental omega-3 fatty acid intake as a cancer prevention strategy in humans.

### Limitations of the Study

While our studies are compelling in regard to the attenuation of PAH-mediated lung carcinogenesis by EPA/DHA, a limitation of this study that we carried out is that the tumorigenesis studies were undertaken at the 120-day time point wherein we could only see lung adenomas after PAH exposure. In the future, we will conduct studies at later time points (e.g., 270 days), when we expect to see adenocarcinomas that are closer to human lung cancers.

## Figures and Tables

**Figure 1 ijms-25-03781-f001:**
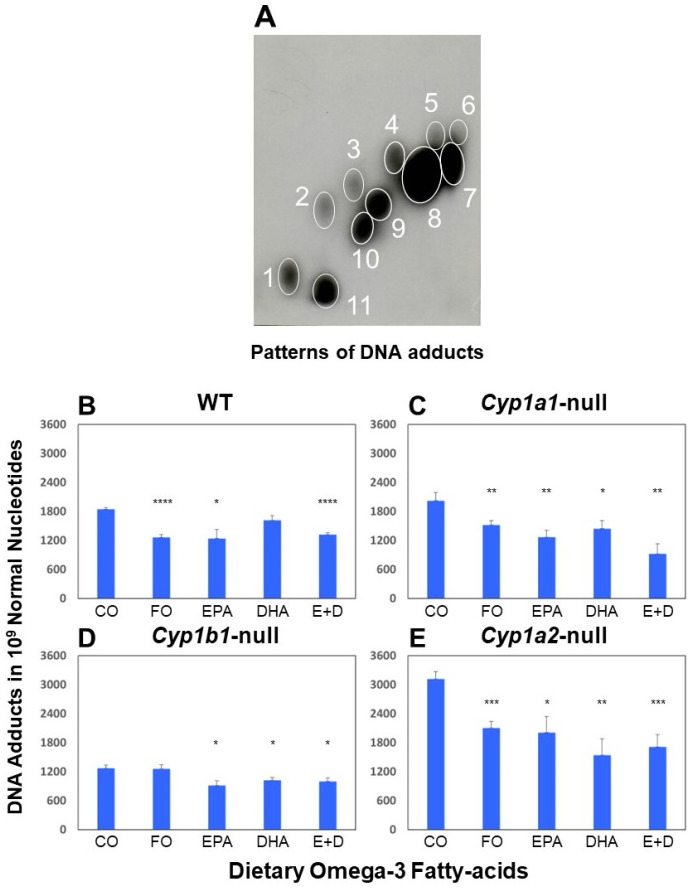
Attenuation of DNA adduct formation by dietary omega-3 fatty acids. Panel (**A**) displays typical pattern of MC-induced DNA adducts detected by ^32^P-postlabeling assay in lungs of WT A/J mice treated with PAH MC (25 mg/kg). Animals were terminated 7 days after MC (25 mg/kg) treatment. Comparison of the total DNA adducts levels in lungs of WT (Panel (**B**)), *Cyp1a1*-null (Panel (**C**)), *Cyp1b1*-null (Panel (**D**)), and *Cyp1a2*-null (Panel (**E**)) mice treated with 25 mg/kg MC after feeding for 30 days with special corn oil (CO) (5%), FO (5%), EPA (750 mg/kg), DHA (500 mg/kg), or EPA + DHA (750 + 500 mg/kg) diets are shown. Animals were euthanized 7 days after MC (25 mg/kg) treatment and lung tissues were collected. Mean ± SEM, n = 4. One way ANOVA analysis was used. * *p* < 0.05; ** *p* < 0.01; *** *p* < 0.005; **** *p* < 0.001.

**Figure 2 ijms-25-03781-f002:**
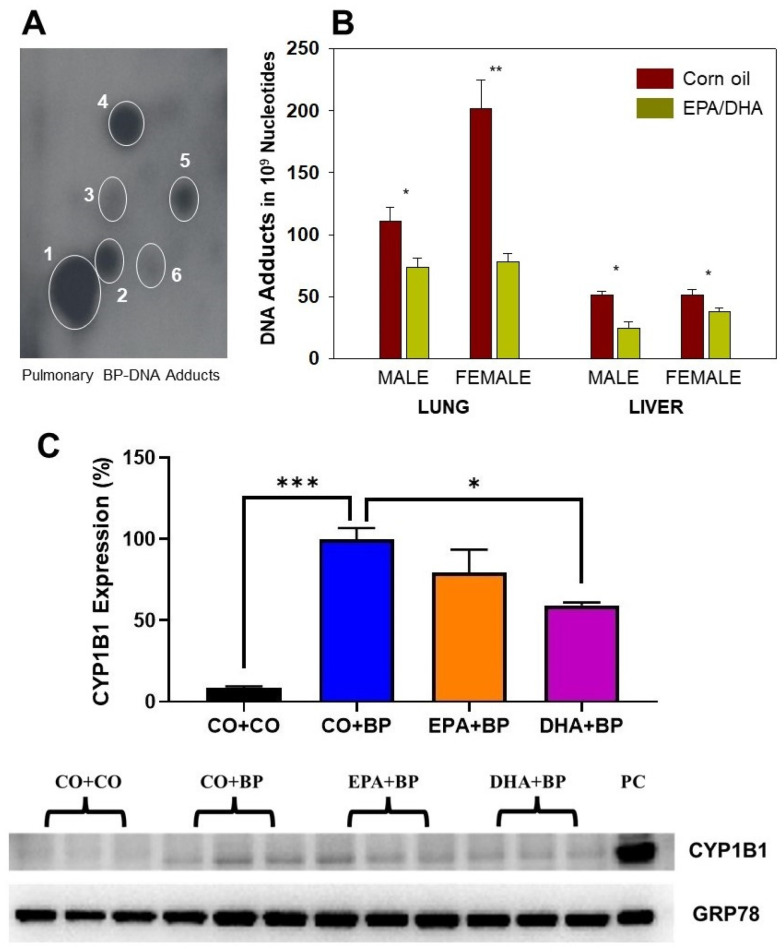
The roles of EPA/DHA and CYP1B1 in the formation of BP-DNA adducts. Panel (**A**): representative autoradiogram of a TLC map showing BP-DNA adducts (mainly derived from BPDE) in lungs of male mice treated with BP. Panel (**B**): EPA/DHA significantly decreased levels of BP-DNA adducts in both lungs and livers of male and female A/J mice. A/J mice (male and female, 6–8 wk.) received EPA (60 mg/kg) and DHA (40 mg/kg) by oral gavage from day 1 to day 10. Meanwhile, mice were treated with BP (10 mg/kg) by i.p. on day 3. Liver and lung samples (n = 4) were collected on day 10 (7 days after BP treatment) for BP-DNA adduct measurements. Panel (**C**): EPA/DHA inhibits CYP1B1 expression induced by BP in lungs of male mice pretreated with CO, EPA, or DHA, and then treated with BP. Proteins (n = 3) isolated from lung microsomes were probed with CYP1B1 antibody by Western blotting. The CYP1B1 levels were estimated by normalizing with GRP78 antibody. EPA and DHA significantly inhibited CYP1B1expression compared to the CO group. Student’s *t*-test (**B**) and ANOVA test (**C**) were used for statistical analysis. * *p* < 0.05; ** *p* < 0.01; ***, *p*< 0.001. PC is the positive control.

**Figure 3 ijms-25-03781-f003:**
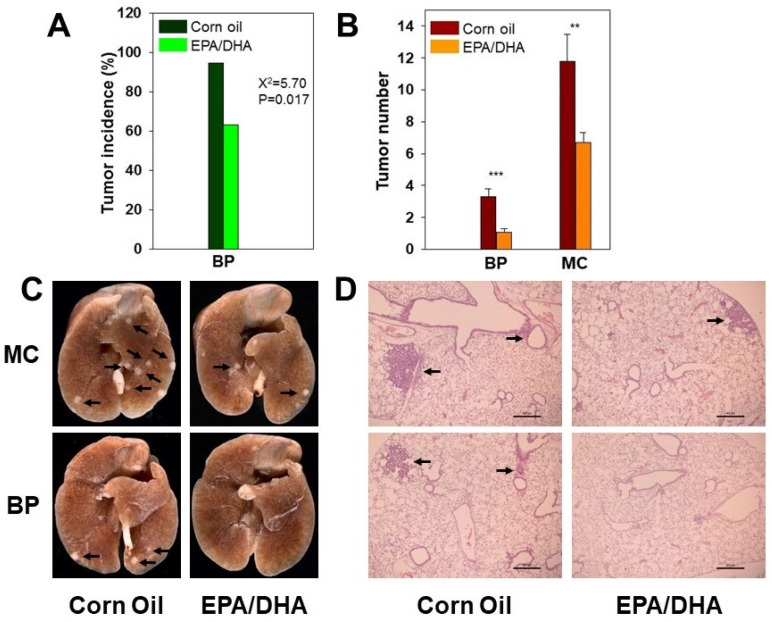
Suppression of PAH-induced lung tumorigenesis by EPA/DHD. A/J mice (male and female, 6–8 wk.) received daily EPA (60 mg/kg) and DHA (40 mg/kg) by oral gavage from day 1 to day 21 and twice/week EPA (60 mg/kg) and DHA (40 mg/kg) by oral gavage from day 21 to day 120. On day 3, mice were treated with BP (60 mg/kg) or MC (10 mg/kg) by i.p. injection for tumorigenesis study. The tumor incidence (Panel (**A**) was 63% in the EPA/DHA + BP group versus 95% in the CO + BP group. The tumor number per animal (Panel (**B**) was also significantly decreased in the EPA/DHA + BP group. While the tumor incidence was 100% in both the MC and MC + EPA/DHA groups, the tumor numbers were significantly attenuated in the MC + EPA/DHA group. Chi-Square analysis (**A**) and Student’s *t*-test (**B**) were used. ** *p* < 0.01; *** *p* < 0.005. Representative lung tissues (Panel (**C**)) from the MC, BP, MC + EPA/DHA, or BP + EPA/DHA groups show significant suppression of lung tumors in the BP or MC + EPA/DHA groups compared to the BP or MC alone groups. Histological staining shows EPA/DHA inhibiting lung tumorigenesis by BP or MC (Panel (**D**)). The animals were treated with BP, MC, or BP + EPA/DHA, or MC + EPA/DHA as described in Figure 2. Lungs were fixed with formalin and processed for histological analyses and stained with H&E. The arrows point to tumorigenic cells (adenomas). The scale bars indicate 400 µm.

**Figure 4 ijms-25-03781-f004:**
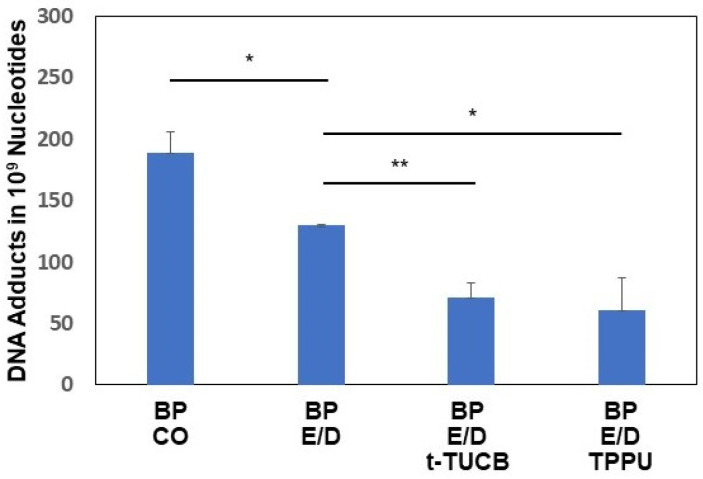
Effects of omega-3 fatty acids and sEH inhibitors on BP-DNA adducts are shown. Male A/J mice (8–10 wk.) were pre-treated with EPA (60 mg/kg)/DHA (40 mg/kg), EPA (60 mg/kg)/DHA (40 mg/kg) + sEH inhibitor t-TUCB (1.5 mg/kg), or EPA (60 mg/kg)/DHA (40 mg/kg) + TPPU (1.5 mg/kg) for three days, and then animals were treated with BP (60 mg/kg) by i.p. Mice continued to receive EPA/DHA or EPA/DHA + sEH inhibitor for an additional day until euthanized. BP-DNA adducts were analyzed by ^32^P-postlabeling. Quantification of total adduct spots are expressed as number of adducts in 10^9^ nucleotides. The data are mean ± SEM of values from 3–4 mice per group. One way ANOVA was used for statistical analysis. * *p* < 0.05; ** *p* < 0.01. E/D stands for EPA/DHA.

**Figure 5 ijms-25-03781-f005:**
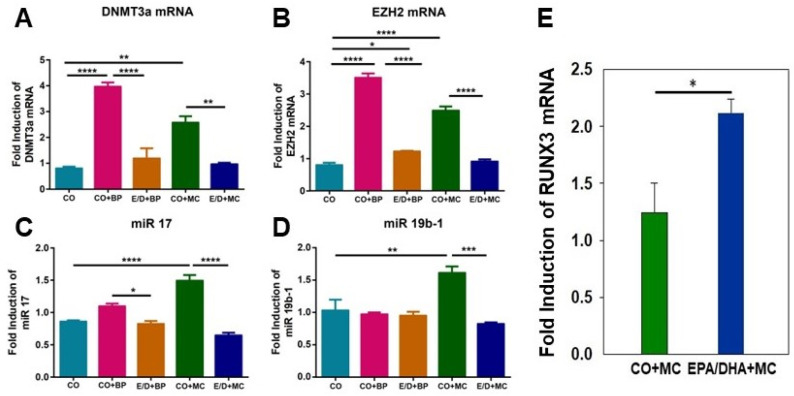
Modulation of DNMT3a, EZH2, miR17 and miR 19b-1, and RUNX3 by PAHs and EPA/DHA. BP and MC induced each of these genes, i.e., DNMT3a, EZH2, miR17, and miR 19b-1, and these genes were suppressed by EPA (E) and DHA (D). (Panels (**A**–**D**)). Panel (**E**) displays the effect of omega-3 fatty acids EPA and DHA on the PAH-mediated induction of pulmonary *Runx3* gene expression in A/J mice. EPA/DHA treatment led to increased *Runx3* expression when compared to PAHs. The animals were treated with BP, MC, BP + EPA/DHA, or MC+ EPA/DHA, as described in Section 4.4.2. Data represent mean ± SEM of 3 individual mice. The symbols *, **, *** and **** represent statistical significance at *p* < 0.05, *p* < 0.01, *p* < 0.005 and *p* < 0.001.

**Figure 6 ijms-25-03781-f006:**
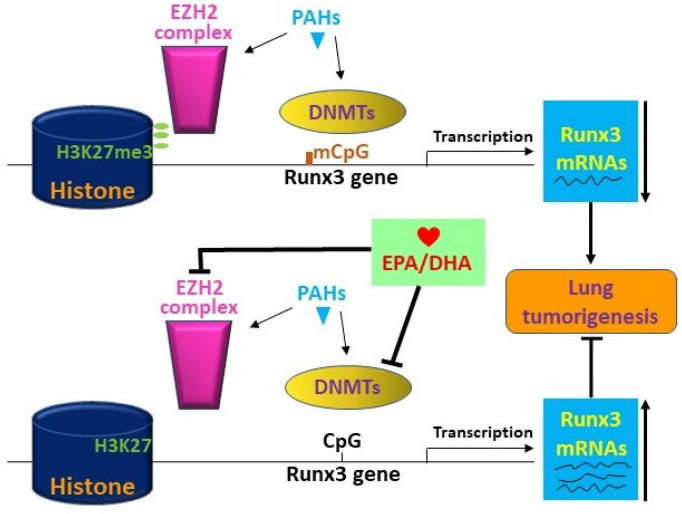
The proposed diagram depicts the mechanisms by which EPA/DHA protect mice against lung tumorigenesis via the RUNX3 gene. We hypothesize that PAHs contribute to lung tumorigenesis in part by inducing DNMT1/3a/3b genes, which will in turn lead to the augmentation of the methylation of the RUNX3 tumor suppressor gene at CpG islands on the promoter. PAHs also elicit the induction of EZH2, which is part of the PRC2 complex that trimethylates H3K27me3. This mechanism in turn attenuates RUNX3 transcription, resulting in the silencing of the RUNX3 gene. We further hypothesize that EPA/DHA will inhibit DNMT1/3a and EZH2, and this result will lead to restoration of expression of RUNX3 and, thereby, attenuation of tumorigenesis.

**Table 1 ijms-25-03781-t001:** Diet component (%).

Ingredients	CO	FO	EPA	DHA	EPA + DHA
Dextrose	54.06	54.06	54.06	54.06	54.06
Casein	26.35	26.35	26.35	26.35	26.35
DL-Methionine	0.34	0.34	0.34	0.34	0.34
Corn Oil (CO)	5.00	0	4.99	4.99	4.75
Fish Oil *	0	5.00	0	0	0
EPA	0	0	0.075	0	0.08
DHA	0	0	0	0.05	0.05
Mineral mix, AIN-76 (Harlan Teklad)	3.91	3.91	3.91	3.91	3.91
Vitamin mix, AIN 76-A (Harlan Teklad)	1.12	1.12	1.12	1.12	1.12
Choline bitartrate	0.22	0.22	0.22	0.22	0.22
Pectin	9.00	9.00	9.00	9.00	9.00

* Fish oil contains a total of 32.2% omega-3 fatty acids including 15.5% EPA and 9.1% DHA.

**Table 2 ijms-25-03781-t002:** The primers used in the real-time PCR.

Gene Name	Forword	Reverse
Runx3	ACCACGAGCCACTTCAGCAG	CGATGGTGTGGCGCTGTA
DNMT1	CCA AGC TCC GGA CCC TGG ATG TGT	CGA GGC CGG TAG TAG TCA CA G TAG
DNMT 3a	CGACCCATGCCAAGACTCACCTTCCAG	AGACTCTCCAGAGGCCTGGT
Ezh2	CTAATTGGTACTTACTACGATAACTTT	ACTCTAAACTCATACACCTGTCTACAT
miRNA 17	TGTCAAAGTGCTTACAGTGCAG	GCATAATGCTACAAGTGCCCTC
miRNA 19b-1	AGTTTTGCAGGTTTGCATCC	CCACCACAGTCAGTTTTGCAT

## Data Availability

All data presented in this study have been shown as Figures in this article.

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
