# Peer review of "Attenuation of Polycyclic Aromatic Hydrocarbon (PAH)-Induced Carcinogenesis and Tumorigenesis by Omega-3 Fatty Acids in Mice In Vivo"

_ijms, 2024, doi:10.3390/ijms25073781_

Round 1

Reviewer 1 Report

Comments and Suggestions for Authors

Comments and Suggestions for Authors

In this study, the authors tested the hypothesis that the omega-3 fatty acids EPA and DHA will attenuate PAH-DNA adducts and further decrease lung tumor incidence and/or multiplicity in mice treated with the PAH benzo[a]pyrene (BP) or 3-methylcholanthrene (MC). They also evaluated some of the mechanisms involved in the attenuation of chemical carcinogenesis and tumorigenesis by omega-3 fatty acids EPA and DHA. Overall, the manuscript addresses an interesting research question and presents valuable data; with the suggested improvements, it has the potential to make a significant contribution to the field. Therefore, I recommend a major revision for the manuscript. I explain my concerns in more detail below. I ask that the authors to specifically address each of my comments in their response.

Abstract: 

Lines 17 – 18: In this study, we tested the hypothesis that the omega-3 fatty acids EPA and DHA will attenuate PAH-DNA adducts and further decrease tumor incidence and/or multiplicity.

The toxicological process is centered on this hypothesis; therefore, the hypothesis of this work is not clear.

Lines 20 – 21: the effects of omega-3 fatty acid on PAH-20 mediated lung carcinogenesis and tumorigenesis was studied; 

It should state: the effects of omega-3 fatty acid on PAH-20 mediated lung carcinogenesis and tumorigenesis were studied;

Introduction section 

Lines 32 – 34: A total of 238,340 new cases of lung cancer and 127,070 lung cancer deaths are projected to occur in the United States in 2023 [1]. Lung cancer is the leading cause of cancer death globally, with an estimated 1.8 million deaths (18%) in 2020 [2].

These data are confusing based on the date of publication. In the discussion section a different reference is used with data in 2021.

Lines 37 – 42: What is the difference between primary and typical sources of PAHs? The sources of PAH are repeated or very similar. The consumption of these PAHs would be more related to another type of cancer, not lung cancer. On the other hand, although environmental factors are the main source, it should be emphasized that these would occur mainly in densely populated areas or areas dependent on the socio-economic context (burning of materials).

Line 48: what does "etc." mean? Specify. It would be interesting to mention under what circumstances these PAHs promote the development of cancer in such diverse organs.

Lines 55 – 56: Reducing the potential development of this type of cancer would be a better term used than "inhibiting it."

Line 64: The authors already measured PAH-DNA adducts in the same model or animals; therefore, the authors should emphasize the originality of this study.

Line 69: “Most cancers are still not curable but are preventable”. 

However, in lines 55 – 56 the authors mention that under this mechanism the development of cancer can be inhibited.

Lines 78 – 79: The authors should consider that the decrease or inhibition in the activity of these CYP1 could cause some negative effect on the metabolism of other xenobiotics, or at least mention it in the discussion of the results.

Lines 84 – 86: This hypothesis is very general and it has been previously tested; therefore, this study lacks a research hypothesis.  

The context of the introduction does not seem appropriate since it appears that only adenomas and not adenocarcinomas were identified in the long-term experiment. Furthermore, the authors do not mention or include the context or the importance of the epigenetic genes in their study model.

Results section 

Line 93: What is the purpose of administering the omega-3 fatty acid diets a month prior to lung carcinogenesis induction with the carcinogen MC?

Lines 101 – 102: Interestingly, further attenuation of pulmonary DNA adducts (up to 57.68%) was observed in Cyp1a1-null mice (Figure 1C) by dietary …

This is correct only for DHA and D + E groups, not for all groups of Cyp1a1-null mice.

Lines 106 – 108: Interestingly, Cyp1a2-null mice displayed the highest levels of pulmonary DNA adducts among the four genotypes of mice (Figure 1E), although CYP1A2 is liver specific, suggesting that hepatic CYP1A2 plays key role of detoxification of MC.

On the contrary, hepatic CYP1A2 play a minor role in the levels of pulmonary DNA adducts since other CYP1 family members participate in the activation of MC compounds in Cyp1a2-null mice. In this sense, what do the authors mean by the detoxification role of MC by CYP1A2? 

Figure 1A: what do the numbers in Figure 1A mean?

Figure 1B – E: The Y-axis scale of graphs B to E should be the same for comparison purposes.

Line 127: … benzo(a)pyrene-7,8-dihyrodiol-9,10-epoxide deoxy-guanosine (BPDE-dG), …

It should say “benzo(a)pyrene-7,8-dihydrodiol-9,10-epoxide deoxy-guanosine (BPDE-dG)”,

Line 128: What do the authors mean by “pure” EPA/DHA? According to line 373, dietary EPA and DHA were purchased.

Lines 134 – 135: In Figure 2A, the authors should specify the sex of the mice analyzed in the autoradiogram. What do the numbers in Figure 2A mean?

Lines 136 – 138: Why do the authors use a completely different experimental protocol in Figure 2A to determine BP-DNA adducts as compared to that of MC-DNA adducts in Figure 1? Omega – 3 pretreatment times? EPA/DHA doses? 

Lines 139 – 143: In Figure 2C, the authors should specify the sex of the mice analyzed by Western blotting.

Lines 159 – 160: Why are the values for MC-treated groups missing in Figure 3A? The authors should clarify this lack of data.

Line 172: The tumor multiplicity or number of tumors per animal (Panel B). 

According to the definition, these terms cannot be used interchangeably, or what do the authors mean by “number of tumors per animal”?

Lines 177 – 178: The authors should indicate or describe in the Results section the type of histological features of lung tumorigenesis found in Figure 3D. Are there any adenocarcinoma or adenomas identified by histology analysis? What is the classification used in the study?

Lines 180 – 181: The arrows are missing in Figure 3D.

Line 83: Typical patterns of pulmonary BP-adducts were observed (Figure 3A). This is incorrect.

Lines 183 – 185: What do the authors mean by “Mice treated with CO, EPA/DHA, EPA/DHA + t-TUCB or TPPU [36] showed similar profiles of DNA adducts”, …

According to Figure 4, the level of DNA adducts is quite different among experimental groups.

Lines 185 – 186: What do the authors mean by “sEH inhibitor t-TUCB (1.5 mg/kg) or TPPU (1.5 mg/kg) did not induce additional DNA adducts”. According to the activation of BP, the product of the EH enzymatic reaction shows a higher pro-mutagenic activity that the precursor; therefore, EH inhibition will not produce additional DNA adducts.

Lines 188 – 189:  What do the authors mean by “suggesting that higher levels of epoxy omega-3 fatty acids suppressed the formation of DNA adducts”?

Lines 191 – 195: Why do the authors use a completely different experimental protocol in Figure 4 to determine BP-DNA adducts as compared to that of MC-DNA adducts in Figures 1 and BP-DNA adducts in Figures 2 and 3? Days after BP treatment? 

Lines 222 – 223: The animals were treated with BP, MC, 222 or BP+EPA/DHA or MC+ EPA/DHA as described in Fig. 2.

However, this is incorrect; since in panel E the authors state “Immunohistochemistry assay further confirmed that MC induced nuclear EZH2 expression levels in tumor cells of mice (Panel E)”; according to the Experiment 2 protocol, in order to obtain tumor samples, animals were sacrificed at day 120. 

Based on the multi-stage carcinogenesis model, it is not clear why these epigenetic genes are measured at different stages of the cancer model and presented in one figure, since molecular evidence indicates that different genes are up- or down-regulated according to the progress of the disease. 

Line 225: Please statistically justify the use of the one-way ANOVA analysis for the comparison of only two population samples in Figure 5G.

Discussion section 

Lines 255 – 256: Same commentary as that in Lines 106 – 108.

Lines 276 – 279: Same observation as that in Lines 106 – 108. In addition, the authors should considerer in their discussion on this matter the higher contribution of phase 2 enzyme conjugation in the liver compared to that of the lung tissue.

Lines 289 – 290: Same commentaries as those in Lines 222 – 223.

Lines 328 – 331: Same observation as that in Lines 185 – 186. EH is involved in the conversion of the more active diol of BP, which is a better substrate for CYP1B1 that the precursor, thus contributing or possible being the main mechanism associated to the suppressed formation of DNA adducts.

This study lacks information regarding its limitations.

Materials and methods section

Lines 382 – 418: Same commentaries as those in Lines 191 – 195.

If female mice showed a higher pulmonary DNA-adducts and these were more effectively decreased by EPA/DHA intervention, why female mice were not included o even selected for the other animal experiments?

Why do the authors use different experimental protocols through the 4.4. Animal Experiments section? Sex, EPA or/and DHA pretreatment period, route of administration, EPA or/and DHA dose or concentration, day of BP treatment, dose of BP, day of sacrifice after BP treatment? 

Line 398: Please define multiplicity or tumor number and provide the findings of the histological analyses. In this regard, the context of the study does not seem appropriate since it appears that only adenomas and not adenocarcinomas were identified in the long-term experiment.

Conclusions section

Lines 469 – 470: Our studies demonstrate that omega-3 fatty acids can significantly attenuate lung carcinogenesis and tumorigenesis in mice exposed to carcinogenic PAHs…

The authors should provide the evidence of lung carcinogenesis in order to state this conclusion. 

The conclusion section would benefit from a discussion of the limitations or potential confounding factors of the experiments conducted.

Reviewer 2 Report

Comments and Suggestions for Authors

This is a very interesting manuscript that addresses the complex steps of carcinogenicity and tumorigenicity and develops a mechanistic platform for the beneficial actions of long chain omega-3 PUFAs, all in an in vivo model.

Minor Issue

It would be useful to explain the rationale for the two different modes of oral administration of EPA and DHA (supplemented in diet, oral gavage).   What was the diet composition used during the gavage experiments?  Might the measurement of plasma fatty acids be helpful for assuring that the modes of oral administration are comparable, if that is the intent?  

The following suggestions are offered to increase the clarity of the manuscript for readers.

Figure 1B-D.  Setting the y-axis range to the same values (0-3600) for all four panels will facilitate a more direct comparison of responses in each mouse model.

Figures 1A:  Alter label to “Pattern of MC-Induced DNA Adducts”, thus reinforcing the known differences in adduct maps among different carcinogens. This would be appreciated by readers unfamiliar with the assay and differences in adducts derived from various carcinogenic compounds.

Line 130:  Given that Experiment 1 utilized WT and three knockout strains, it would be helpful to the reader if it is stated explicitly that the WT mouse is used in subsequent experiments (#2, ~Line 130 and #3, ~Line 154).

Figure 2B: Asterisks appeared to be misaligned with the data bars.

Figure 2C: Should the y-axis label read “CYP1B1 protein expression” or “relative expression”?  Inhibition suggests measurement of enzymatic activity.

Figure 3C &D:  Addition of arrows indicating tumors (3C) and adenomas (3D) would be helpful for the reader.  The legend suggests arrows are present, but they are not visible.

Line 187: ‘decreased’ could be a better word choice than ‘exacerbated’ 

In future studies comparable to Expt2, would it be possible to collect tissues for the measurement of EPA, DHA and there oxylipin metabolites?   Highly sensitive mass spec techniques are now readily available for the latter.  

Figure 5:  As currently presented, Panels A-F are too small the read. Panel G: Neither the text, figure legend nor graph make it clear what the method of RUNX3 detection is (protein, IHC, RNA?) 

Can the methylation status of the RUNX3 gene be assessed in the context of future studies in order to continue to test the hypothesis developed based in the findings in this manuscript?

Please review the references for consistency of format and check PubMed for some very recent published papers on sEH inhibitors that could be relevant to the manuscript.

Reviewer 3 Report

Comments and Suggestions for Authors

The authors of this manuscript investigated the effect of the omega-3 fatty acods EPA and DHA on the PAH-DNA adducts formation and the lung tumor incidence and/or multiplicity in mice treated with the PAH benzo[a]pyrene (BP) or 3-methylcholanthrene (MC). Results of the study show that omega-3 fatty acids can substantially attenuate lung carcinogenesis and tumorigenesis in mice treated with carcinogenic PAHs via decreased PAH-DNA adducts production, downregulation of CYP1B1 expression, inhibition of genes associated with epigenetic regulation and protection agains lung tumorigenesis through the RUNX3 gene. It is suggested that omega-3 fatty acids may be a potential chemo-preventive agent for lung cancer. Future studies are required for the investigation of the efficacy and safety of supplemental omega-3 fatty acid intake as a cancer prevension strategy. 

The paper was well prepared with scientifically sound introduction, appropriately chosen and exectuted methodology, as well as discussion adequately comparing so far knowledge of other scientistswith the obtained results. 

A few specific comments that might be considered:

1. Animal groups are characterized by a small number of individuals (n=4) and I am concerned about the statistical tests chosen by the authors. All the more so because the information about the statistics used is unclear. Please standardize captions under figures -For example: First -  Provide the title of the figure, then short description of panels and clear description of statistics, eg. "All values are shown as mean ± SEM (n = 5); Two-way ANOVA with post-hoc Tukey test: *p < 0.05; **p < 0.01; ***p < 0.001". what program was used for the analysis and graphics? For example "The statistical analysis was performed using Prism 8 (GraphPad Software)".

2. Figure 2C - It would be more transparent if the X axis is written as simply "CYP1B1 expression", because now the message of the graph is unclear - whether treatment with BP inhibits or induces CYP1B1 the most. Moreover, the student's t-test chosen for statistics is incorrect - the student's t-test is used to compare two groups. Please do the statistics using the ANOVA test and provide a correct description of the figure.

3. Figure 3, lines 167-181 - there is no information about the statistical tests used in Fig3A and 3B and about the number n taken into account in the graphs.

4. 2.4 Inhibition of sEH enhances EPA/DHA-mediated prevention of pulmonary carcinogenesis, line 183. Figure 3A shows the tumor incidence in groups treated with BP, not typical patterns of pulmonary BP-adducts. Please correct

5. Figure 4 lacks the information about what statistical test was performed

6. Line 204 - to study the effect of PAHs on the expression og the genes associated with epigenetic regulation in the presence and absence itd. The sentence is unclear in the present form.

7. Figure 5 should be more legible. Ponadto proszÄ™ poprawić opis wg wczeÅ›niejszych wskazówek - Title, e.g. DHA/EPA diet causes reduced expression of genes involved in carcinogenesis. Effect of BP or MC and E/D diet on the expression of A) DNMT3, B) EXH2, C) miR17 and D) miR 19b-1. Results are presented as mean +/- SEM, n=3-4; Two-way ANOVA with appropriate post-hoc test: *p < 0.05; **p < 0.01; ***p < 0.001". E) Induced nuclear EXH2 expression levels mediated by MC in mice tumor cells evaluated immunohistochemically and F) suppressed in mice treated with EPA/DHA +MC. G) The effect of EPA and DHA on PAH-mediated induction of pulmonary Runx3 gene expression in A/J mice.

8. It would be more transparent if the discussion would start with the most important discoveries od the work - the beginning of the discussion (Lines 244-248) fits more with the introduction.

9. Line 374 delete extra space

Table 1 - standardize that everywhere, e.g. 2 or everywhere 3 decimal places

Line 395 - delete extra space before "For"

Line 413 - delete extra space before "or" 

10. Instead "epigenetic genes" pleas use the phrase "genes associated with epigenetic regulation".

Round 2

Reviewer 1 Report

Comments and Suggestions for Authors

Suggestions for Authors

I greatly appreciate the authors' corrections to the various comments made. In reply to some of your responses, I suggest the following recommendations, if you consider it necessary.

Introduction section 

Suggestion: Line 50: I do know what "etc." means; I was referring to specify, according to scientific evidence, those organs in which tumors develop due to exposure to PAHs for better knowledge of the reader, or it can be indicated that these are the main organs affected by exposure to PAHs instead of "etc.".

Original concern: Lines 57 – 58: “Therefore, a decrease in PAH-DNA adduct levels is expected to diminish or inhibit tumor formation”.

Response: Yes. Decrease in carcinogen DNA adducts can reduce the chance of tumor formation or tumor incidence.

Suggestion: Therefore, a decrease in PAH-DNA adduct levels is expected to diminish or reduce the potential development of this type of cancer.

Original concern: The context of the introduction does not seem appropriate since it appears that only adenomas and not adenocarcinomas were identified in the long-term experiment. Furthermore, the authors do not mention or include the context or the importance of the epigenetic genes in their study model.

Response: We agree that only adenomas were identified, but these adenomas could develop into adenocarcinomas at later time points. We have stated this in the Introduction. We also discussed the role of genes involved in epigenetics in tumorigenesis. For example, Runx3 is a tumor suppressor gene which is induced by EPA/DNA (Figure 5), suggesting that EPA/DHA can in part suppress tumorigenesis by augmenting runx3 gene.

Suggestion: I was referring to the fact that in the Introduction section the authors do not include information about the importance of the epigenetic regulation in their study model, since, as they actually commented in their response, the results in Figure 5 address this potential mechanism of action by EPA/DHA. Again, the introductory section does not indicate this line of research in its study model and yet it was already introduced in the key words.

Results section 

Original concern: Line 96: What is the purpose of administering the omega-3 fatty acid diets a month prior to lung carcinogenesis induction with the carcinogen MC?

Response: The reason for the month-long diet was to determine the impact of dietary omega-3 fatty acids on PAH-mediated lung carcinogenesis.

Suggestion: However, this is a primarily preventive strategy that may not apply or be relevant, especially in high-risk populations, such as smokers and other people exposed to environmental carcinogens and whose therapies suggest another type of approach or treatment.

Conclusions section

Original concern: Lines 469 – 470: Our studies demonstrate that omega-3 fatty acids can significantly attenuate lung carcinogenesis and tumorigenesis in mice exposed to carcinogenic PAHs…

The authors should provide the evidence of lung carcinogenesis in order to state this conclusion. 

Response: Omega-3 fatty acids significantly attenuated DNA adduct levels in lung of four genotypes (Fig. 1, Fig. 2B) and tumor incidence (Fig. 3A) and tumor number (Fig. 3B to 3D). This is adequate evidence that omega 3 fatty acids attenuated lung carcinogenesis.

Suggestion: “Our studies demonstrate that omega-3 fatty acids can significantly attenuate lung tumorigenesis in mice exposed to carcinogenic PAHs…
